# Non-Pharmacological Interventions for Wandering/Aberrant Motor Behaviour in Patients with Dementia

**DOI:** 10.3390/brainsci12020130

**Published:** 2022-01-19

**Authors:** Tatiana Dimitriou, John Papatriantafyllou, Anastasia Konsta, Dimitrios Kazis, Loukas Athanasiadis, Panagiotis Ioannidis, Efrosini Koutsouraki, Thomas Tegos, Magda Tsolaki

**Affiliations:** 11st Department of Neurology, Aristotle University of Thessaloniki, 44 Salaminos Street, Halandri, 15232 Athens, Greece; 23rd Age Center IASIS, 2nd Neurology Department, University of Athens, ‘Attikon’ Hospital, 73 Krimeas str., Glyfada, 16562 Athens, Greece; jpapatriantafyllou@gmail.com; 31st Department of Psychiatry, “Papageorgiou” General Hospital of Thessaloniki, Aristotle University of Thessaloniki, 54124 Thessaloniki, Greece; konstaa@auth.gr (A.K.); loukatha@outlook.com.gr (L.A.); 43rd Neurology Department, Aristotle University of Thessaloniki, 54124 Thessaloniki, Greece; dimitrios.kazis@gmail.com (D.K.); ekoutsou@auth.gr (E.K.); ttegos@auth.gr (T.T.); tsolakim1@gmail.com (M.T.); 52nd Department of Neurology, Aristotle University of Thessaloniki, 54124 Thessaloniki, Greece; ispanagi@auth.gr

**Keywords:** wandering, aberrant motor behaviour, BPSD, dementia, RCT

## Abstract

Background: Aberrant motor behaviour or wandering refers to aimless movement without a specific purpose. Wandering is common in patients with dementia and leads to early institutionalization and caregivers’ burden. Non-pharmacological interventions should be also considered as a first-line solution for the wandering because current pharmacological treatment has serious side-effects. Methods: A cross-over randomised controlled trial (RCT) with 60 participants of all stages and different types of dementia was conducted in Greece. The sample was randomly assigned in 6 different groups of 10 participants each. Every intervention lasted for 5 days, and there were 2 days as a wash-out period. There was no drop-out rate. The measurements used were the Mini Mental State Examination (MMSE), Addenbrooke’s Cognitive Examination Revised (ACE-R), Geriatric Depression Scale (GDS), Functional Rating Scale for Symptoms in Dementia (FRSSD), and Neuropsychiatric Inventory (NPI). The interventions that were evaluated were reminiscence therapy (RT), music therapy (MT), and physical exercise (PE). Results: NPI scores were reduced in the group receiving PE (*p* = 0.006). When MT (*p* = 0.018) follows PE, wandering symptoms are reduced further. RT should follow MT in order to reduce wandering more (*p* = 0.034). The same combination was effective for the caregivers’ burden as well; PE (*p* = 0.004), MT (*p* = 0.036), RT (*p* = 0.039). Conclusions: An effective combination that can reduce wandering symptoms in all stages and types of dementia was found: The best order was PH-MT-RT. The same combination in the same order reduced caregivers’ burden.

## 1. Introduction

Dementia is a term that is used for a range of diseases that cause cognitive impairment, most often in the elderly, resulting in difficulty in performing everyday activities [1]. According to the World Health Organization, dementia is estimated to affect nearly 78 million people by 2030 and more than 150 million by 2050 [2]. A total of 90–97% of dementia patients will develop behavioural and psychological symptoms of dementia (BPSD) as the disease progresses [3]. BPSD can cause a decrease of cognitive ability, an increase of caregiver burden, and hospitalization [3]. Among BPSD, aberrant motor behaviour or wandering is one of the most challenging behaviours.

Wandering is defined as a pointless pursuit in a state of disorientation [4]. It is a syndrome of dementia that is related to motor behaviour. The patient has frequent, repetitive disorientation and/or spatially disorientation that is manifested in random pacing patterns, some of which are associated with elopement attempts or getting lost when alone [4]. Some other studies identified that sometimes wandering has purpose. When a patient with dementia (PwD) has become lost in the community, it is often described as “wandering” [5]. This behaviour has been found to occur in 65% of patients with Alzheimer’s disease (AD) [6]. Previous studies have underlined this challenging behaviour as a major problem for the caregivers because wandering may lead patients to do harm, to fall, and to have exposure to extreme weather conditions if he/she wanders outdoors [4,7]. Many patients with dementia (PwD) wander with no orientation to time and place [4]. This is critical because they can be exposed to malnutrition, dehydration, and sleeping problems [8]. In some extreme cases, wandering may also lead to death. In addition, the patient may sometimes be agitated while wandering and hurt himself/herself [4]. Wandering may be increased when the environment is unfamiliar [5]. Wandering may be a way to escape from an unpleasant situation, such as isolation, loneliness, boredom, or frustration [9]. It is a very critical behaviour because, according to the literature [10], if a PwD, who has been lost has not been found within 24 h, up to half of those patients will suffer from injuries or death.

The aetiology of wandering remains unclear. It seems that three fields have been examined in order to understand the aetiology of this behaviour. These fields are biological, psychosocial, and environmental [4].

According to the biological hypothesis, the impairment of the brain functions is detected in spatial memory, visuospatial ability, or executive functions [11]. Wandering is related to visuospatial impairments, but spatial memory also plays a major role in this behaviour [12]. Optic flow perception and interpretation lead to spatial navigation problems, which is the basis for wandering in PwD [13]. Decision-making, planning, and executive impairments also lead to wandering. In terms of neuropathology, wanderers have a more severely reduced cerebral blood flow in the left temporoparietal region [14]. PET scans also show decreased frontotemporal glucose utilization and dopamine [13]. There are also circadian rhythm problems in wanderers, and therefore, many wanderers have sleeping disturbances [13]. Overall, it seems that wandering involves parietal and frontal dysfunctions and possibly temporal impairments as well [13]. A spatial and executive impairment potentially leads to wandering.

According to the psychosocial and environmental approach, wandering can occur because the patient feels discomfort, or the environment is unpleasant or noisy [15]. PwD may need to go to the bathroom, and they get frustrated and disoriented in the space, and therefore, they wander. Others may feel unsafe, so they seek out a familiar face or place, and therefore, they may wander [14]. Personality may also play a role, as previous studies have shown. PwD who are sad or angry may sit alone in their rooms for long periods and then wander [4]. People with dementia who express their feelings may wander more.

The current treatment for wandering is limited. There is limited evidence of the efficacy of the pharmacological treatment [4]. It is also crucial that the current medicine lead to a higher risk of aggravation and mortality. Currently, antipsychotics are prescribed to control wandering symptoms. However, the side effects of antipsychotics should be considered [16]. The antipsychotic drugs may cause severe extrapyramidal side effects (EPS), such as muscle rigidity, tremor, bradykinesia, and akathisia. They block dopamine D2 receptors in the striatum [17]. Bradykinesia refers to reduced motor activity, which may lead to akinesia in several cases [18]. Tremor affects the hands, feet, and head, and it is a rhythmic muscle contraction. Another side effect is an increased muscle tone (rigidity) and a slow gait [17]. In addition, akathisia refers to restlessness and repetitive movements of the feet [17]. PwD who suffer from akathisia cannot keep sitting and therefore shift body position. Akathisia is a side effect that frequently occurs after starting antipsychotics or after increasing the dose of the drug [18]. Moreover, dystonia causes muscle contraction and attacks the tongue, trunk, limbs, and the neck muscles [18]. Although antipsychotic drugs are widely used to treat BPSD, with a prescription rate of 20–50%, except for haloperidol and risperidone in some countries, other antipsychotic drugs are not approved to treat BPSD [19]. Therefore, they are prescribed as off-label. Apart from the EPS, the antipsychotics also cause agitation, psychosis, aggression, and inappropriate behaviours [18]. Furthermore, risperidone and olanzapine demonstrated an increased risk of cardiovascular effects [20]. According to some previous studies [21,22,23], there is increased risk of mortality in PwD who are treated with antipsychotics than in PwD who do not take any antipsychotic drug. Additionally, cholinesterase inhibitors lead to a reduction of wandering, but they also have serious side effects [24]. Therefore, non-pharmacological interventions should be a first-line treatment. Currently, global positioning system (GPS)s are widely used for wanderers. Other non-pharmacological interventions include blocking areas or locking doors but are related to ethical issues (such as loss of independence and dignity) [25]. Environmental modifications, door camouflaging, door alarms, and exercise seem to have some positive results.

The aim of the current study is to find a combination of non-pharmacological interventions that can effectively reduce wandering in PwD and reduce caregivers’ burden as well. The first objective of the current study is to compare different combinations of strategies in PwD.

## 2. Methods

In the current study, we included sixty patients (N = 60) with dementia and wandering symptoms from the Neurological Department of the General Hospitals of Thessaloniki and Athens. The participants were randomly assigned to 6 different groups of 10 participants each. There were no criteria for the randomization of the participants in order to avoid risk of bias. The inclusion criteria were (a) suffering from dementia, (b) having wandering symptoms according to the Neuropsychiatric Inventory (NPI), and (c) with caregivers who were eager to cooperate. The study is in accordance with ethical principles (declaration of Helsinki). Their mean age was 73.40 (SD 8.86). The criteria for the diagnosis of AD and MCI due to AD is in accordance with the National Institute of Neurological and Communicative Disorders and Stroke (NINCDS) and the Alzheimer’s Disease and Related Disorders Association (ADRDA) [26]. The patients were diagnosed with Alzheimer’s disease (AD), vascular dementia (VD), Lewy body dementia (DLB), frontotemporal dementia (FTD), mild cognitive impairment (MCI), mixed type (AD and VD), and AIDS. The mean years of education WERE 9.57 (SD 4.63). In all, 60% of the sample (N = 36) were females. The demographic characteristics of the sample are shown on Table 1.

### 2.1. Procedure

This is a randomized, controlled, crossover trial. The NPI inventory applied to the family caregivers before any treatment and guidelines. The interventions lasted for five days, and there were two days as a wash-out period. The interventions were applied every morning after breakfast because it was the common hour for the caregivers, and everyone agreed to the schedule. NPI questionnaire was applied before any intervention (baseline NPI), and on the morning of the sixth day, it was applied again in order to evaluate the results of the interventions. The comparisons were made between NPI baseline and the results of the sixth day of each week.

### 2.2. Interventions

#### 2.2.1. Reminiscence Therapy (RT)

RT is used to recall past memorable events of the patient’s life. Photos, music, books, and letters can be used. The caregiver aims to involve the patient in a discussion and/or evaluation of the past experiences and events. In accordance with previous studies [27,28], the current trial used photo albums in order to recall happy days. The intervention lasted 60 min per session (as most of the previous studies applied their interventions for 60 min) [27] and took place once a day after breakfast for five days.

#### 2.2.2. Music Therapy (MT)

MT is a non-pharmacological intervention that has been proven very effective for the reduction of some BPSD, such as depression [29] and agitation [30]. MT has not been examined as an alternative method for wandering symptoms. However, because previous studies have claimed that MT has a beneficial and promising effect on BPSD, the intervention was used in the current study [31]. The caregivers chose the most preferable music for their patients. In this study, MT was demonstrated for 45 min once per day (as most of the previous studies applied their interventions for 45 min) [29] for five days a week, every morning after breakfast.

#### 2.2.3. Physical Exercise (PE)

PE has shown very promising results for the reduction of many BPSD [32,33]. For that reason, it was used in the current study as well. Our intervention was administrated every day for 30 min every morning after breakfast (as most of the previous studies applied their interventions for 30 min) [32]. All the family caregivers chose walking, as it is the easiest physical activity for their patients.

Each group received the three aforementioned interventions in a different sequence. Given that RT is intervention A, MT is intervention B, and PE is intervention C, the sequence was as follows: Group 1 received the interventions in the sequence ABC, group 2 ACB, group 3 BAC, group 4 BCA, group 5 CAB, and group 6 CBA. The sequence of the procedure is shown in Table 2.

### 2.3. Measures

Mini Mental State Examination (MMSE) [34,35]: MMSE is a 30-point questionnaire that is used to evaluate the cognitive status. It is used to estimate the severity of cognitive decline. The questionnaire examines registration, attention, recall, language, praxis, and orientation. Higher scores indicate better cognitive performance.

Addenbrooke’s Cognitive Examination Revised (ACE-R) [36,37]: ACE-R is a 100-point questionnaire that is used to evaluate the cognitive impairment. It includes MMSE. It is highly sensitive and can be used to support the diagnosis of dementia. Higher scores indicate better cognitive performance.

Geriatric Scale of Depression (GDS) [38,39]: This scale is a questionnaire of 30 YES/NO questions that examines if the patient has depression. Higher scores indicate higher level of depression.

Functional Rating Scale for Symptoms in Dementia (FRSSD) [40,41]: It is a scale to access the activities of daily living. The scale is a questionnaire that is completed by caregivers and includes 14 different daily activities, such as eating, dressing, incontinence, speaking, sleeping, recognition of faces, personal hygiene, memory of names, memory of events, alertness, agitation, spatial orientation, emotional status, and socialization. The scale is scored from 0–3 for each question (where 0 = fully independent, and 3 = fully dependent). Higher scores indicate lower level of functionality (activities of daily living).

Neuropsychiatric Inventory (NPI) [42,43]: This questionnaire is given to the caregiver. It evaluates the frequency and severity of the symptoms and the impact of each behaviour on the caregiver. The domain total score is the product of (a) frequency multiplication severity score and (b) the total score of the caregiver’s distress. A total score is obtained by summing all the domain total scores. It is a flexible, easy to use, and a valid and reliable tool. It evaluates a wide range of psychopathology, including hallucinations, and it can be used across different ethnic groups [44]. In the current study, only the sub questions that refer to wandering/aberrant motor behaviour were used.

### 2.4. Data Analysis

Categorical variables were presented as percentages, while continuous variables were presented as mean value and standard deviation (SD). Wilcoxon signed-rank test was used because the distribution of the differences between the samples cannot be assumed to be normally distributed. Chi-square test was used to find differences in gender in the six groups, and finally, z-value score was used in order to find the type of dementia in each group. *p*-Values less than 0.05 were considered statistically significant. SPSS 25.0 (IBM Inc., Armonk, NY, USA) was used for the statistical analysis.

## 3. Results

The mean scores of all the patients for all measurements used are as follows: MMSE 18.22 (SD 5.07), ACE-R 55.87 (SD 19.15), GDS 8.03 (SD 4.3), FRSSD 17.33 (SD 9.35), and NPI 7.82 (SD 1.87). Results are shown on Table 1. Table 3 refers to the percentage of the different dementia types of the sample. A total of 60% of the sample suffered from AD, 5% from VD, 5% from DLB, 5% from PDD, 10% from FTD, and 15% from mixed dementia (AD and VD). According to the results, there is an effective combination that can reduce wandering in PwD, which is physical exercise, followed by music therapy, followed by reminiscence therapy. Group 6 had the most effective results. In the first week, PE was applied, and this intervention reduced wandering significantly from the baseline NPI score (*p* = 0.006). The second week, MT was applied, further reducing the behaviour of wandering (*p* = 0.018). The third week, after MT, RT was applied, and it reduced wandering even more (*p* = 0.034). The mean score of the caregivers’ burden because of wandering was also reduced after the first week of PE intervention (*p* = 0.004). MT applied in the second week reduced the burden further (*p* = 0.036). In the third week RT was applied, it reduced the burden even more (*p* = 0.039). Table 4 and Table 5 show the results analytically.

## 4. Discussion

The current results reveal that there is a combination of non-pharmacological interventions that can reduce wandering in PwD. The combination of the interventions is PE, followed by MT, followed by RT. The study found that this combination can effectively reduce wandering symptoms in all types and different stages of PwD. According to the results, this combination is also effective in reducing the caregivers’ burden due to wandering. Some other combinations seemed to work as well; however, the above-mentioned sequence of the interventions offered the best results. In particular, in group 4, MT, followed by PE, followed by RT seem to have some beneficial effects, and group 5 showed some positive results when PE was followed by RT and MT applied last (Table 4 shows the results).

According to our knowledge, no other RCT has been conducted to find efficient combinations of non-pharmacological interventions for wandering. One study from Korea [45] examined the effect of doll therapy (a recognised intervention to help address cases of agitation and depression in PwD). Dolls were proven to provide a source of comfort or meet the patient’s need to care for someone else in 51 PwD, and this study found that this intervention can effectively reduce wandering, among other negative behaviours (aggression, obsessions, negative verbalizations, etc.). However, no significant details on the methodology were provided in the current study.

PE (in the current study, the caregivers used walking) is an intervention with beneficial effects on other BPSD as well [46]. More expressive personalities who wander decompress themselves via walking [47]. To our knowledge, no other RCT has examined the effect of walking on the wandering symptom. MT is the intervention that follows PE in our combination. MT has been shown to be a promising intervention for the treatment of many BPSD [31]. No other RCT has examined MT for wandering. Finally, RT is another non-pharmacological intervention that has shown positive results for the treatment of the BPSD [28]. To our knowledge, however, no other RCT has examined its effect in wandering.

The combination that effectively reduced wandering in PwD was the same combination that effectively reduced caregivers’ burden. the caregiver’s psychology is strictly related to the patient’s behaviour. The caregivers look for a way to reduce the symptom, its frequency, or its severity. Therefore, because the above combination reduces wandering, its frequency, or its severity, it is effective for the caregivers’ burden as well.

Future studies should focus on the non-pharmacological interventions for the treatment of BPSD. The studies of non-pharmacological interventions are still in early stages. There is a great need for research on the current field. We need to expand the interventions, to simplify the process of the interventions, and to find better ways in order to effectively reduce the behavioural symptoms, the needs, and the difficulties that occur in every stage of the disease. Multiple types of research designs can give us better results. At present, non-pharmacological interventions are not funded, and there are no organizations that invest in testing them. It is crucial to have in mind that the combinations of the interventions may give better results than one intervention by itself. Therefore, different methods of administering an intervention may benefit the PwD at different stages of dementia. Another critical matter is the different cultural backgrounds of the PwD and their caregivers. Therefore, the combinations of the interventions should be applied in accordance with the cultural background of the patient.

The hypothesis that there is a combination of non-pharmacological interventions that can effectively reduce wandering in PwD has been confirmed.

## 5. Strengths and Limitations

The strengths of this study are the sample size, the randomization of the groups, and the methodology. The study used common non-pharmacological interventions, which have been shown to be beneficial in other BPSD, in order to examine their effectiveness for reducing wandering symptoms. No other RCT has been conducted so far that has focused only on wandering or has found a combination of non-pharmacological interventions that can effectively reduce wandering and caregivers’ burden.

On the other hand, the limitations of this study are that the interventions lasted for five days, there was an absence of follow-up, and the interventions were performed by informal caregivers, and there was no control group. The study meant to help the caregivers find fast and efficient solutions for wandering, which is why the interventions took place for a short period of time. The caregivers were unprofessional, however, so they had strict and rigorous instructions from the clinicians, and they were able to speak to a specialist at any moment of the trial. Finally, due to costs, GPS machines could not be used even though the literature refers to their related beneficial outcomes.

There is a strong need for further research and more RCTs to find efficient solutions for wandering in dementia. There are non-pharmacological interventions that can effectively reduce wandering and caregivers’ burden, and they need more research to identify effective combinations of interventions.

## 6. Conclusions

In sum, this study found an effective combination of non-pharmacological interventions that can reduce wandering in PwD and caregivers’ burden as well. The combination applies to both genders, all stages, and different types of dementia. The study suggests that common non-pharmacological interventions, which are beneficial for other BPSD, should also been examined for wandering. The interventions that were used in the current study are cost-effective and can be applied by informal caregivers as well. To our knowledge, no other RCT has been conducted for non-pharmacological treatment for wandering symptoms in PwD. Future studies should examine wandering more, as it is a behaviour with many crucial consequences to the PwD and their caregivers.

## Figures and Tables

**Table 1 brainsci-12-00130-t001:** Baseline characteristics of the sample.

	Mean (SD) or N (%)
**Females, N (%)**	60% (N = 36)
**Age**	73.40 (8.86)
**Years of education**	9.57 (4.63)
**MMSE**	18.22 (5.07)
**ACE-R**	55.87 (19.15)
**GDS**	8.03 (4.3)
**FRSSD**	17.33 (9.35)
**NPI Results**	7.82 (1.87)
**NPI Distress**	3.43 (0.76)

**Table 2 brainsci-12-00130-t002:** Sequence of the procedure, A = Reminiscence therapy (RT), B = Music therapy (MT), C = Physical exercise.

Sequence	1st Week	2nd Week	3rd Week
ABC	A	B	C
ACB	A	C	B
BAC	B	A	C
BCA	B	C	A
CAB	C	A	B
CBA	C	B	A

**Table 3 brainsci-12-00130-t003:** Percentages of the different types of dementia of the sample.

AD	VAD	LBD	PDD	FTD	Mixed (AD&VAD)
60%	5%	5%	5%	10%	15%

**Table 4 brainsci-12-00130-t004:** Results of dementia patients.

Group 1	NPI original	NPI before A	A–B	B–C
Mean score ± SD	8 ± 1.26	8 ± 1.26–8 ± 1.26	8 ± 1.26–8 ± 1.26	8 ± 1.26–4 ± 1.41
Percentiles	-	6–8.25, 6–8.25	6–8.25, 6–8.25	4–6.50, 4–6
*p*	-	1	1	0.007
**Group 2**	**NPI original**	**NPI before A**	**A–C**	**C–B**
Mean score ± SD	8 ± 2.35	8 ± 2.35–8± 2.35	8 ± 2.35–4 ± 1.79	4 ± 1.79–8 ± 2.35
Percentiles	-	6–9, 6–9	6–9, 4–6.50	4–6.50,6–9
*p*	-	1	0.007	0.757
**Group 3**	**NPI original**	**NPI before B**	**B–A**	**A–C**
Mean score ± SD	8 ± 1.88	8 ± 1.88–8 ± 1.88	8 ± 1.88–8 ± 1.88	8 ± 1.88–6 ± 1.03
Percentiles	-	6–8.25, 6–8.25	6–8.25, 6–8.25	6–8.25, 4–6
*p*	-	1	1	0.031
**Group 4**	**NPI original**	**NPI before B**	**B–C**	**C–A**
Mean score ± SD	8 ± 1.88	8 ± 1.88–8 ± 1.88	8 ± 1.88–4 ± 1.03	4 ± 1.03–8 ± 1.88
Percentiles	-	5.50–8.25, 5.50–8.25	5.50–8.25, 4–6	4–6, 5.50–8.25
*p*	-	1	0.012	0.511
**Group 5**	**NPI original**	**NPI before C**	**C–A**	**A–B**
Mean score ± SD	8.5 ± 1.85	8.5 ± 1.85–6 ± 1.33	6 ± 1.33–8.5 ± 1.85	8.5 ± 1.85–8.5 ± 1.85
Percentiles	-	8–9.75, 5.50–6.50	5.50–6.55, 8–9.75	8–9.75, 8–9.75
*p*	-	0.025	0.500	1
**Group 6**	**NPI original**	**NPI before C**	**C–B**	**B–A**
Mean score ± SD	8 ± 1.68	8 ± 1.68–5 ± 1.39	5 ± 1.39–4.5 ± 1.68	4.5 ± 1.68–3 ± 1.68
Percentiles	-	7.50–9, 4–6	4–6, 3–4	3–4, 2–2
*p*	-	0.006	0.018	0.034

**Table 5 brainsci-12-00130-t005:** Results for the caregivers.

Group 1	NPI original	NPI before A	A–B	B–C
Mean score ± SD	3 ± 0.73	3 ± 0.73–3 ± 0.73	3 ± 0.73–3 ± 0.73	3 ± 0.73–2 ± 0.56
Percentiles	-	2–3.25, 2–3.25	2–3.25, 2–3.25	2–3.25, 1.75–2
*p*	-	1	1	0.110
**Group 2**	**NPI original**	**NPI before A**	**A–C**	**C–B**
Mean score ± SD	3.5 ± 0.52	3.5 ± 0.52–3.5 ± 0.52	3.5 ± 0.52–1.5 ± 0.82	1.5 ± 0.82–3 ± 0.87
Percentiles	-	3–4, 3–4	3–4, 1–2.25	1–2.25, 2–4
*p*	--	1	0.004	0.366
**Group 3**	**NPI original**	**NPI before B**	**B–A**	**A–C**
Mean score ± SD	4 ± 0.84	4 ± 0.84–3 ± 0.78	3 ± 0.78–4 ± 0.84	4 ± 0.84–2 ± 0.56
Percentiles	-	3–4, 3–4	3–4, 3–4	3–4, 1.75–2
*p*	-	0.022	0.102	0.006
**Group 4**	**NPI original**	**NPI before B**	**B–C**	**C–A**
Mean score ± SD	3 ± 0.91	3 ± 0.91–2 ± 0.66	2 ± 0.66–1.75 ± 2.25	1.75 ± 2.25–3 ± 0.91
Percentiles	-	3–4, 2–4	2–4, 1.75–2.25	1.75–2.25, 3–4
*p*	-	0.309	0.028	0.410
**Group 5**	**NPI original**	**NPI before C**	**C–A**	**A–B**
Mean score ± SD	4 ± 0.67	4 ± 0.67–2 ± 0.78	2 ± 0.78–4 ± 0.67	4 ± 0.67–3.5 ± 0.99
Percentiles	-	3.75–4, 1.75–3	1.75–3, 3.75–4	3.75–4, 2–4
*p*	-	0.004	0.942	0.063
**Group 6**	**NPI original**	**NPI before C**	**C–B**	**B–A**
Mean score ± SD	4 ± 0.67	4 ± 0.67–2 ± 0.42	2 ± 0.42–1.5 ± 0.73	1.5 ± 0.73–1 ± 0.37
Percentiles	-	3–4, 1.75–2	1.75–2, 1–2	1–2, 1–1
*p*	-	0.004	0.036	0.039

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
