# Peer review of "Non-Pharmacological Interventions for Wandering/Aberrant Motor Behaviour in Patients with Dementia"

_brainsci, 2022, doi:10.3390/brainsci12020130_

Round 1
Reviewer 1 Report
This is a cross-over randomized controlled trial on different treatment or intervention approaches to treat wandering or aberrant motor behaviors in people diagnosed with dementia. The manuscript is well-written, the topic is of interest for the readers, and the findings are really interesting in the field of dementias. However, several minor changes should be addressed prior to the publication of results.
Abstract. In the methods section, in the abstract, the authors describe that they carried out a RCT. This abbreviation should be previously mentioned. This may also apply for all the measurement strategies they are reporting: MMSE, ACE-R, GDS, FRSSD, NPI. I can understand that there is few lines to do that, but this is crucial to understand the study. In the results section of the abstract, it is reported " PE can reduce NPI score from baseline". The proper way to describe it should be: "NPI scores were reduced in the group receiving PE". This is a result, not an interpretation. "Can reduce" should be reomved out.
Introduction. The introduction section is really brief. The summary about pharmacological treatment of wandering/ aberrant motor behavior should be expanded. I consider that the main argument to study non-pharmacological interventions should be drawn after revising pharmacological interventions and highlighting the side-effects of medications in that cases.
The main aim of the current study was to find a combination of non-pharmacological interventions to reduce wandering. However, the first objective may be to compare different strategies or a combination of strategies in dementia patients.
Methods. How were patients randomized? How have the authors decided which 10 patients were assigned to each intervention?
In the methods section the authors describe the three interventions but they reported the patients were assigned to one of the 6 groups. This should be clarified.
The results section is really brief. I would expand it and divide results into subsections.
Table 1 is mentioned in the third line, followed by Table 3 (Table 3 refers...). Where is Table 2 mentioned?
Results should also report measures (or scores) by each group (from the six groups).
The discussion section is brief. It should be rewritten and compared it with similar trials in dementia. Although they report that no other RCT has examined the effects of waking on the wandering symptoms, this should be better discussed. Combination of treatments should be also discussed in depth and the authors should propose new studies or future research on the topic.
Author Response
Please see the attachment, with yellow are the corrections that you have suggested.
1) Abstract: line 30: was rephrased as asked 2) in the abstract in overall we have changed the abbreviations to the full titles (such as Mini Mental State Examination instead of writing only MMSE) 3) line 41 was rephrased as asked 4) the introduction has been updated with more information on wandering and the current pharmacological treatments 5) the objective of the study has been updated in the final paragraph of the introduction 6) in the methods we have made it clear how the randomisation of the process has been made 7) line 156 we have explained why the interventions took place after the breakfast 8) line 159 we have explained the comparisons of the NPI 9) in the interventions we have explained why we used the duration of time that it was said 10) we have added a paragraph in order to explain the sequence of the process 11) we have mentioned table 2 and the more analytically results in table 4 and 5 12) line 248 we have explained other combinations that have been found to be quiet effective 13) we have added in the discussion section what future studies should do 14) we have added as a limitation to our study the lack of control group 15) we have uploaded all our tables
Reviewer 2 Report
Line 30- The authors mention that there are no previous RCTs using these three therapies alone or in combination, thus this bold statement should be tempered until more evidence confirms efficacy. Non-pharmacological therapies should be considered as an initial strategy to manage wandering. Rephrase sentence.
Line 34 Authors need to convey that the three therapies were provided serially, as it first appears each group only experienced each type of therapy. The concept of testing different orders of therapy did not come across in the abstract until Line 42. This needs to be clarified.
Lines 61 and 63 (as well as throughout the manuscript). Authors should replace roman numerals references with numbers instead of mixing styles. Please refer to current issues of Brain Science and instructions to the authors for correct manuscript reference formatting. Also the Bibliography should be uniform which it is not currently.
Line 126 -In the methods, authors need to convey that different serial combinations of therapies were evaluated. Describe the various combinations (were all combinations attempted, if so, this needs to be conveyed..the missing tables may have helped to clarify this)
Tables 1, 2, 3 are missing. Table 2 is not mentioned in text, but Tables 1 & 3 are, yet not uploaded with manuscript.
There are different times for each therapy 60, 45, 30 minutes how was that determined, and explain why the morning was decided at the period to perform the therapies.
Line 179- This study design involves repeated measures, and yet this is not taken into account in the data analysis section, consultation with a statistician is needed.
Line 194 identifies Group 6 as most effective, I assume the missing tables describe results from the other groups? Clarify the groups.
Lines 195-201. Were comparisons at the end of each week of therapy compared to baseline, or the scores of the previous week of therapy. Clarify comparisons.
Line 208 identify the other combinations that “seemed to work” perhaps from most to least effective.
Lines 221-225. Need to clarify that “walking alone” or MT alone or RT alone has not been evaluated in RCTs. It is mentioned that these therapies have been shown to be effective in impacting wandering, but if there are no RCTs, then what type of evidence supports their selection for this study. This needs to be clarified in the introduction, and were the durations of therapy in the current study similar to those used in past studies?
It is possible that just greater (or focused) one-on-one caregiver attention (for 30, 45, 60min) everyday would have similar positive impact on wandering regardless of a named therapy? This could be addressed in the limitation section as there were no control groups.
Author Response
Please see the attachment (green are the changes that you have suggested)

Round 2
Reviewer 2 Report
The authors of this manuscript have adequately addressed this reviewer's comments. I have no further suggestions.